# Development of Alzheimer’s Disease Biomarkers: From CSF- to Blood-Based Biomarkers

**DOI:** 10.3390/biomedicines10040850

**Published:** 2022-04-05

**Authors:** Sakulrat Mankhong, Sujin Kim, Seongju Lee, Hyo-Bum Kwak, Dong-Ho Park, Kyung-Lim Joa, Ju-Hee Kang

**Affiliations:** 1Department of Pharmacology, Research Center for Controlling Intercellular Communication, College of Medicine, Inha University, Incheon 22212, Korea; sakulratkulrat@gmail.com (S.M.); sujin2419@inha.ac.kr (S.K.); 2Program in Biomedical Science and Engineering, Inha University, Incheon 22212, Korea; lees@inha.ac.kr (S.L.); kwakhb@inha.ac.kr (H.-B.K.); dparkosu@inha.ac.kr (D.-H.P.); 3Department of Anatomy, College of Medicine, Inha University, Incheon 22212, Korea; 4Department of Kinesiology, Inha University, Incheon 22212, Korea; 5Department of Physical & Rehabilitation Medicine, College of Medicine, Inha University, Incheon 22212, Korea; drjoakl@inha.ac.kr

**Keywords:** Alzheimer’s biomarker, cerebrospinal fluid, blood-based biomarker, clinical trial, amyloid-beta, tau

## Abstract

In the 115 years since the discovery of Alzheimer’s disease (AD), our knowledge, diagnosis, and therapeutics have significantly improved. Biomarkers are the primary tools for clinical research, diagnostics, and therapeutic monitoring in clinical trials. They provide much insightful information, and while they are not clinically used routinely, they help us to understand the mechanisms of this disease. This review charts the journey of AD biomarker discovery and development from cerebrospinal fluid (CSF) amyloid-beta 1-42 (Aβ42), total tau (T-tau), and phosphorylated tau (p-tau) biomarkers and imaging technologies to the next generation of biomarkers. We also discuss advanced high-sensitivity assay platforms for CSF Aβ42, T-tau, p-tau, and blood analysis. The recently proposed Aβ deposition/tau biomarker/neurodegeneration or neuronal injury (ATN) scheme might facilitate the definition of the biological status underpinning AD and offer a common language among researchers across biochemical biomarkers and imaging. Moreover, we highlight blood-based biomarkers for AD that offer a scalable alternative to CSF biomarkers through cost-saving and reduced invasiveness, and may provide an understanding of disease initiation and development. We discuss different groups of blood-based biomarker candidates, their advantages and limitations, and paths forward, from identification and analysis to clinical validation. The development of valid blood-based biomarkers may facilitate the implementation of future AD therapeutics and diagnostics.

## 1. Introduction

After the first case description of Alzheimer’s disease (AD) in 1906 as ‘presenile’ dementia, there has been immense and significant progress in the understanding, awareness, diagnosis, and treatment of this condition. The definitive diagnosis of AD is characterized by two pathologic hallmarks at postmortem, amyloid β (Aβ) deposition in senile plaques and tau deposition neurofibrillary tangles (NFT). The diagnosis of AD is based on clinical symptoms, cognitive investigation, and other etiologies of dementia. While the measurement of cerebrospinal fluid (CSF) biomarkers (i.e., amyloid β1-42, total tau, and phosphorylated tau) indicate disease status, Aβ positron emission tomography (PET), and tau PET imaging tests can monitor the pathologic changes in living AD patients [1,2]. According to the Aβ cascade hypothesis, accumulation of aggregated forms of Aβ induces neurotoxicity and degeneration; therefore, the measurement of CSF Aβ1-42 levels or quantification of deposition by Aβ PET in living persons were proposed as proxies for AD [3,4]. The use of imaging and CSF biomarkers provides objective evidence for the underlying neuropathology of cognitive impairment to clinicians and researchers, even though these techniques have not been implemented in routine clinical practice. In addition, the features of CSF AD biomarkers are very closely correlated with those measured in imaging modalities, including magnetic resonance imaging (MRI) and PET [5]. Very recently, in 2018, the latest guidelines from the National Institute on Aging and the Alzheimer’s Association (NIA-AA) set out a research framework approach towards AD diagnosis, which consists of Aβ deposition (A), tau pathology (T), and neurodegeneration or neuronal injury (N) [6]. The number of current blood-based biomarkers has expanded and is closely related to practicality using the recent ATN criteria as a framework [7]. In parallel, technical developments have made great progress in the measurement of these biomarkers. There is a great need for well-timed and precise AD diagnosis because it goes beyond providing diagnostic and prognostic information about the disease to both the clinician and patient. An even more critical step will be providing appropriate care and treatment strategies or opportunities to join clinical trials. The underlying etiology and spatiotemporal pattern (i.e., stage of disease and region of pathology) are difficult to determine without biochemical and imaging biomarkers (Figure 1); therefore, the development of AD biomarkers to support the clinical diagnosis is as essential as a therapeutics strategy. Under the NIA-AA research framework of ATN classification, single biomarker evidence of abnormal Aβ alone would be biologically assigned as “Alzheimer’s pathologic change”; biomarker evidence of abnormal tau pathology and/or neurodegeneration without Aβ abnormality would be assigned as “non-Alzheimer’s pathologic change”. If both abnormal Aβ and tau pathology are observed, the individual can be assigned “Alzheimer’s disease”, and an individual can be biologically assigned as “Alzheimer’s and concomitant suspected non-Alzheimer’s pathologic change” when both Aβ abnormality and neurodegeneration are observed [8]. Therefore, the absence of Aβ abnormality in biomarker analysis can be referred to non-Alzheimer’s neurodegenerative diseases, such as Lewy body dementia and frontotemporal dementia. 

In this paper, we describe the development of AD biomarkers from previous decades to the current day and assess future perspectives, as well as the progression of analytical platforms. In addition, we emphasize the emerging blood-based AD biomarkers that are currently gaining traction, and the correlations among blood-based biomarkers and CSF or imaging biomarkers. The perspective and future direction of AD biomarkers’ development are discussed regarding the implementation of biomarker development and utilization in clinical practice.

## 2. Discovery and Development of AD Biomarkers

### 2.1. History of CSF AD Biomarkers

The development of CSF AD biomarkers over the past three decades provides premortem evidence for amyloid and tau pathologies in the AD brain (Figure 2). In 1984, the National Institute of Neurological and Communicative Disorders and Stroke and the Alzheimer’s Disease and Related Disorders Association (NINCDS-ADRDA), a workgroup on AD diagnosis, initially described the clinical criteria for AD. The criteria for diagnosing AD were categorized into “definite (presence of autopsied pathological evidence)”, “probable (maximum level of antemortem certainty)” or “possible” [9,10]. In 1985, the observation that cerebral amyloid protein forms a plaque core in Alzheimer’s disease was first reported [11]. CSF Aβ was reported in 1992, identifying a vital AD biomarker, although initial reports on CSF Aβ were unclear [12]. Subsequently, in 1993 the first CSF T-tau was measured by sandwich enzyme-linked immunosorbent assay (ELISA) [13]. In 1995, two papers reported the reduction of Aβ42 and tau elevation in CSF of AD patients compared with neurologic controls [3,14]. The core CSF biomarkers, Aβ42, T-tau, and p-tau, have been assessed in abundant studies regarding fluid AD biomarker development. In 2010, Jack et al. proposed a hypothetical model of AD biomarkers in which Aβ biomarkers become abnormal first (i.e., decreased CSF Aβ42 level and/or Aβ PET positivity) sequentially followed by tau pathology (i.e., increased CSF p-tau and/or Tau PET positivity), neurodegenerative biomarkers and cognitive symptoms [15]. Later, the Dominantly Inherited Alzheimer Network (DIAN) results supported the hypothesis that suggested the progressive decline of CSF Aβ42 may become abnormal before an abnormal amyloid PET finding. The DIAN study recruited adult family members of a clinically affected parent with autosomal dominant early onset AD (EOAD) in which the parent has a known mutation causing symptomatic AD. The results from the DIAN study strongly support the suggestion that the genetic causes of EOAD follow a common pathway in AD pathophysiology independent of the underlying initial genetic cause. Although the time taken to reach defined CSF cutoff values for the identification of underlying AD pathologies may be different with sporadic AD, the clinical and biomarker trajectories in EOAD are consistent with sporadic AD [16,17,18]. Additionally, the study suggested that tau becomes abnormal before detection by FDG PET and that FDG PET and MRI reveal abnormalities in close temporal proximity to each other [16]. 

Based on the considerable clinical evidence for the reflection of underlying pathologies in AD by CSF AD biomarkers, the NIA-AA recommended diagnostic guidelines for non-symptomatic (i.e., preclinical) and symptomatic (i.e., mild cognitive impairment and dementia) stages of AD in 2011 by the incorporation of AD biomarkers into the diagnostic criteria for research [9,19,20,21]. In 2011, the AD biomarkers were grouped into two categories which include amyloid and tau-related neurodegeneration. However, tauopathy and neurodegeneration/injury were placed in the same biomarker category. Likewise, neurodegeneration/injury have tau pathology without AD conditions [8,9]. Therefore, more recently, in 2018, NIA-AA revised the diagnostic criteria and provided a new diagnostic approach based on the ATN classification system for AD biomarkers [8]. These criteria form a research framework organized around biomarkers by measuring distinct pathologic processes rather than the measurement modality (e.g., imaging, biofluids). Three components of biomarkers for AD neuropathologic changes are grouped into Aβ deposition (A), tau pathology (T), and neurodegeneration or injury (N). Considering the invasiveness in measurement of CSF AD biomarkers, the validity of blood AD biomarkers is much more acceptable in the clinical setting. Despite disadvantages such as non-specificity to brain pathologies or dilution of marker molecules, blood-based biomarkers have been developing to become a new practical approach in the future (see below).

Neurons cannot survive without the support of glial cells; therefore, several candidates of AD biomarkers in CSF and blood that originated from glial cells have been suggested (Figure 2). For example, astrocytes in AD are reactive playing roles in Aβ clearance. Astrocyte AD biomarkers including GFAP [22,23,24], S100B [24,25] and YKL-40 [26,27] were proposed. However, at this point, no additional clinical advantages of these astrocyte AD biomarkers compared to the “core” AD biomarkers have been proved [28]. Since microglia is closely linked to amyloid and tau pathology, neuroinflammation, and synaptic loss [29], several biomarkers originating from microglia have also been proposed. As one of the AD risk genes, triggering receptor expressed on myeloid cells 2 (TREM2) is mainly expressed on the surface of microglia. Cleavage of TREM2 by metalloproteinase releases soluble TREM2 (sTREM2), which is increased in AD continuum, compared to normal control [30,31,32,33]. Therefore, sTREM2 in biofluid can be a potential AD biomarker that reflects microglial activation in the AD brain. Although the results are controversial, CX3CL1 (also known as fractalkine) and progranulin (PRGN) were proposed as other microglial AD biomarkers. The changes in CX3CL1 throughout the stages of AD are dynamic, reflecting the roles of microglia in AD progression. Since AD brain has low lysosomal and autophagic capacity [34], elevation of PRGN in AD may be an early event to prevent further progression of neuropathology [35]. However, the clinical relevance of glial AD biomarkers should be further evaluated, and the association of glial activation with ATN criteria remains to be elucidated.

Considering the heterogeneous pathologic features of AD, a significant number of AD patients have one or more concomitant pathologic findings, including Lewy bodies, vascular pathologies, TDP-43 inclusion and hippocampal sclerosis [38]. When the AD-associated biomarkers with other pathologic protein signatures are combined, the diagnostic performance of core CSF AD biomarkers is improved. For example, inclusion of measures of mismatch between CSF α-syn and p-tau181 could improve the diagnostic performance of core CSF AD biomarkers and better predict longitudinal cognitive changes [39]. Therefore, the implementation of methods detecting non-amyloid blood biomarkers for the improvement of clinical performance of ‘core’ blood AD biomarkers and for the possible classification of heterogeneous AD patients is needed. This effort will further refine the biological definition of Alzheimer’s disease by the NIA-AA research framework. 

### 2.2. Analytical Platforms of Core CSF Biomarkers

The biomarker-based diagnosis of AD can be classified into two categories, namely brain imaging and fluid biopsy. CSF, owing to its direct contact with the extracellular space of the brain, is the most useful biological fluid reflecting molecular events in the brain; hence, research efforts have been made to develop biochemical biomarkers for AD diagnosis. When Aβ is accumulated as a form of insoluble plaque in the brain, the accumulation can be assessed semi-quantitatively by the injection of radioactive PET ligands that selectively bind to Aβ plaque, including ^11^C-labeled Pittsburgh compound B, ^18^F-florbetapir, ^18^F-florbetaben and ^18^F-flutemetamol. The accumulation of Aβ plaque leads to reduced appearance of Aβ, particularly Aβ42, in CSF. Aβ42 concentration in CSF depends not only on the presence of aggregates but also on the total amount of Aβ peptides in the CSF, reflecting the variable efficiency of amyloid precursor protein processing. Therefore, the implementation of Aβ40 to normalize the CSF Aβ42 concentration to the level of total Aβ peptides, using the Aβ42/Aβ40 ratio, is useful [40,41]. Other imaging biomarkers are useful to assess the functional (e.g., ^18^F-FDG-PET for glucose metabolism) and structural changes (e.g., magnetic resonance imaging) in the brain of AD patients and to predict the progression of MCI to AD. However, as compared to CSF AD biomarkers or amyloid PET imaging, the abnormality of ^18^F-FDG-PET or structural changes in the brain can be detected in the later stage of significant neural dysfunction and cell death (Figure 1). The development and analytical and clinical validation of CSF AD biomarker tests and Aβ PET, and more recently Tau PET imaging tests, provide objective evidence for the respective pathologic changes involved in AD. As described above, the single-plex ELISA platform was first applied to measure Aβ42, total tau (T-tau), and phosphorylated tau (p-tau) in CSF for the discrimination of AD from cognitively normal older people or non-AD dementia. Development of ELISA platform that is acceptable analytical performance supports the intended clinical uses [42]. In 2005, the microsphere-based Luminex-xMAP technology (xMAP) with a flow cytometric multiplex immunoassay method was developed, allowing simultaneous measurement of Aβ42, T-tau, and p-tau [43]. The analytical performance and clinical utility of ELISA and xMAP are comparable. However, these ‘*manual*’ immunoassay platforms showed large inter-laboratory variability in the values of biomarkers. The large inter-laboratory variability is caused by pre-analytical and analytical sources of variability, and efforts to standardize these sources of variability have been made by means of multidisciplinary collaboration [44]. Recently, the ‘fully automated’ immunoassay platforms of Elecsys by Roche and Lumipulse system by Fujirebio were introduced. The inter-laboratory analytical variability in CSF AD biomarkers measured by the fully automated platforms was lower than the variability in manual assays [45,46], which may accelerate the determination of universal cut-off values and their clinical application to early diagnosis of AD. In addition, the plausible universal cut-off values can be used for A/T/N classification and clinical reporting for patient care [47], although there is a high variability in the context of the use of CSF tests, interpretation, and reporting methods. Considering the clinical reporting system, a recent consensus for harmonization in the biochemical diagnosis of AD has been published [47]. Of note, the use of fully automated immunoassay platforms can be used to compare the values of CSF AD biomarkers from different races, which may reveal possible inter-ethnic differences in diagnostic cut-off values [48,49]. Here, we will not discuss analytical platforms in detail as the characteristics, pros and cons, and analytical performance (e.g., precision, analytical sensitivity and specificity, and reproducibility) of the analytical methods are discussed elsewhere [42,50,51,52].

## 3. New Diagnostic Approach-Based A/T/N Biomarkers

Breakthroughs in AD biomarkers include the CSF and PET markers of Aβ and tau proteins, which paved the way for the working group’s efforts that proposed the current ATN framework (Figure 2). This classification represents a conceptual framework for diagnosing AD biologically identified by CSF examination or brain imaging such as MRI or PET [6]. Using a binary category (e.g., A+/−), ATN biomarker profiles are represented by seven biomarkers including CSF Aβ1-42, amyloid PET, CSF p-tau, tau PET, CSF T-tau, ^18^F-fluorodeoxyglucose (FDG-PET), and MRI. Regarding core CSF biomarkers, decreased CSF Aβ1-42 reflects Aβ deposition or Alzheimer’s pathology, increased CSF p-tau reflects abnormal tau, and high CSF T-tau indicates neurodegeneration or injury [6]. Although a positive “A” represented Aβ pathology, as depicted by increased amyloid PET uptake, a decreased level of CSF Aβ1-42 was insufficient to diagnose the stage of neuropathological change. Neurofibrillary tangle-related tau has been suggested as the principal-stage AD biomarker, indicating the disease progression level [53]. Consequently, “T” refers to tau pathology represented by positive Tau PET or increased levels of CSF p-tau. Therefore, detecting aberrations in the levels of both CSF Aβ1-42 and p-tau (i.e., A+T+) can define the pathologic state through the Alzheimer’s continuum, i.e., preclinical AD, prodromal AD, and AD with dementia. In addition, CSF biomarkers and PET imaging in both Aβ and p-tau were discordant in some studies because of the techniques used, the laboratory variability, and the biological differences [54]. However, a series of analytical platforms might disentangle this matter [49,55,56]. Lastly, the “N” biomarkers comprise diverse physiological processes closely interconnected to clinical symptoms, defined as elevated CSF T-tau, a decreased signal on FDG-PET, and grey matter atrophy on structural MRI. Although the elevation of CSF T-tau levels indicates synaptic loss and neurodegeneration in AD, the abnormality of CSF T-tau could be attributed to other neuronal injury-associated diseases [2,57]. Therefore, these biomarkers of neuronal injury are not specific to AD. However, the combination of tau biomarkers and neurodegeneration may provide significant evidence of the disease stage (i.e., T+N−) that may imply AD continuum, whereas T−N+ may represent other causes of neurodegeneration. Integrating three components into the ATN scheme will allow emerging diagnostic criteria that can potentially increase etiological homogeneity and the possibility of specifying the probable AD progression individually. Improvement beyond the current ATN biomarkers is required, particularly N biomarkers that help to identify patients with mild cognitive impairment (MCI) who are at a high risk of cognitive decline and distinguish suspected non-Alzheimer’s pathophysiology [58,59,60]. Recently, several studies on blood-based biomarkers for ATN classification have emerged that could facilitate early diagnosis of patients with AD through an easy and minimally invasive approach, reducing costs and providing greater accessibility [61]. We suggest that blood-based biomarkers should be validated and included for alternative ATN classification to provide prospective avenues for AD detection and clinical trials.

## 4. Blood-Based Biomarkers for AD

Although current biomarkers for AD diagnosis are grounded in neuroimaging and CSF assessment, there are some limitations, such as the invasiveness of the lumbar puncture for CSF collection, lower accessibility, and expense of PET imaging. Due to these limitations, blood-based biomarkers for AD have thoughtfully been developed with several advantages, such as being less invasive, more accessible, and more cost-effective. Initially, the development of blood-based biomarkers of AD had various boundaries that needed to be overcome [62,63,64]. For instance, proteins originating in the central nervous system (CNS) must cross the blood–brain barrier to be detected in the periphery. Accordingly, the concentration of target proteins tends to be much lower in blood relative to CSF [65]. Additionally, expressed targeted proteins in the periphery may represent systematic changes rather than pathologic changes in the brain. Moreover, they may undergo proteolytic degradation in plasma and clearance in the liver or kidneys, further reducing the quantity of protein and raising the discrepancy with CSF levels. In addition, a high number of other proteins in the blood, such as albumin, immunoglobulins, autoantibodies, and heterophilic antibodies, may interfere in the immunoassay, whereas these proteins are considerably lower in CSF samples [66]. Initial measurements of blood-based biomarkers gave contradictory findings. For example, in 2016, a meta-analysis revealed that plasma Aβ42 and Aβ40 concentrations were no different between AD and control, whereas plasma T-tau was strongly associated with AD and CSF core biomarkers [67,68]. Several possible reasons might account for this difference, including those mentioned earlier and the analytical performance of the immunoassay platforms. Previous studies using conventional ELISA platforms showed modest concordance between blood Aβ concentrations and CSF biomarkers for AD [67]. However, accumulated evidence and substantial efforts generated new highly sensitive assays and technologies with remarkable consistency across different cohorts and enabled precise analytical performance. For example, immunomagnetic reduction assays quantify plasma Aβ by measuring the percentage of magnetic signal reduced by immunocomplex formation at the surface of magnetic beads [69]. Another platform for ultrasensitive immunoassay techniques, single-molecule array (SIMOA), has enabled the detection of extremely low concentrations of promising AD biomarkers in blood at picomolar levels. SIMOA with digital counting technology has been used to analyze plasma Aβ1-42, T-tau, and p-tau, as well as NfL [70,71]. Mass spectrometry-based assays proved to be superior over other assays, with a good receiver operating characteristic area under the curve (ROC-AUC) for amyloid PET status of 0.80–0.89 compared with around 0.68 for ELISA and 0.40–0.77 for SIMOA assay [72]. Recent plasma Aβ species measured using mass spectrometry coupled with immunoprecipitation have been introduced with a close correlation with CSF Aβ42/40 ratio [73] and amyloid PET [72]. More recently, a high-resolution mass spectrometry (MS)-based blood test for Aβ conformed to the standards of Clinical Laboratory Improvement Amendments (CLIA) [74,75]. In addition, the PrecivityAD™ test for plasma LC-MS/MS assays of Aβ quantification and qualitative APOE isoform-specific prototyping was completed in the first phase of the Plasma test for Amyloid Risk Screening study for clinical use to evaluate individuals experiencing early cognitive impairment [74]. In the meantime, novel assay platforms to discover and measure blood AD biomarkers are emerging. Full achievement of CLIA standards for p-tau have been obtained in phases 1 to 3, with partial achievement for Aβ according to the 5-phase framework “Biomarker Roadmap” for AD [70]. Initially, blood-based tests for AD seemed unreachable, but so far, the rapidly growing effort has made significant progress and led to tests becoming both accurate and reproducible. However, the desire to advance the use of blood-based biomarkers should be validated in clinical practice. In addition, the analytical performance of the blood assays should be defined in the larger clinical samples with pre-analytical and analytical variability sources. This article has reviewed and updated the emerging blood-based biomarkers for AD and correlated each blood-based biomarker with either CSF or amyloid-PET or Tau-PET scans.

### 4.1. Core AD Biomarkers (Aβ, p-Tau, and T-Tau)

Several studies of plasma biomarkers focused on Aβ and compared the correlation between plasma Aβ and CSF Aβ or Aβ PET [48,53,54,76,77]. Plasma total Aβ40 and Aβ42 levels have been suggested to rule out brain β-amyloidosis in preclinical and early-stage AD [76,77]. The plasma Aβ42/40 ratio, especially combined with age and APOE ε4 status, accurately diagnoses brain amyloidosis and has been suggested for use in screening cognitively normal individuals for brain amyloidosis [77]. Plasma MDS-OAβ combined with APOEε4 and age accurately identify brain amyloidosis in a large Aβ-confirmed population relevant to clinical trials [78]. Plasma MDS-OAβ levels could be beneficial for pre-screening in clinical trial settings, as their measurement could potentially reduce costs [78].

Plasma T-tau was recently transferred onto the SIMOA platform, an ultrasensitive measurement in the blood [2]. Fossati et al. suggested that adding plasma tau to CSF tau or CSF p-tau measures might improve diagnostic accuracy [79]. Post-translational modification of tau protein (e.g., phosphorylation, acetylation, or truncation) undergoes misfolding and subsequent formation of soluble toxic oligomers, which propagates the AD pathology. Accumulation of pathologic tau oligomer, likewise Aβ oligomer, is a rather early event in the progression of AD pathology than NFT formation, which warrants the detection of plasma tau oligomer [80]. Recent studies have shifted towards a focus on plasma p-tau, but this is still challenging so far because of its low concentration and difficulty to detect. Pereira and colleague [81] indicated that plasma Aβ42/40 and p-tau217 could be advantageous in clinical practice, research, and drug development as prognostic markers of future AD pathology. Plasma biomarkers of ATN (amyloid-β42/40, p-tau217, and neurofilament light; NfL) showed utility in improving the prediction of cognitive decline in cognitively unimpaired elderly populations [81,82]. Shen and colleagues reported that p-tau181 significantly correlated with brain amyloid and FDG-PET, which may be a predictive biomarker for Alzheimer’s pathophysiology [72]. The plasma level of p-tau181 significantly correlated with CSF p-tau181 in AD dementia, which was first reported using a modified SIMOA T-tau assay [83]. In agreement with this line of evidence, Brickman and colleagues showed that plasma p-tau181, p-tau217, and NfL were associated with pathological and clinical diagnoses, which suggested use in future pre-screening for clinical AD [84]. Among these, plasma p-tau217 had significantly higher diagnostic accuracy for clinical AD than plasma p-tau181 and plasma NfL but showed no significant distinction to CSF p-tau and tau PET [82]. A longitudinal study demonstrated that plasma p-tau217 has the potential to monitor AD progression [82,85]. Recently, the LC-MS/MS analytical platform was shown to accurately identify brain amyloid status based on a single blood sample and showed excellent performance compared with CSF or amyloid PET [75]. Three biomarkers, amyloid, p-tau, and NfL, are close to clinical implementation [7].

Blood biomarkers have become a milestone in AD biomarker development, and new approaches have ensued [7]. Although we are still in the early stages of blood-based AD biomarker development, many studies have reported the correlation between blood-based and core CSF biomarkers and brain imaging (e.g., amyloid PET, tau PET, and FDG PET) (Table 1). Concentrations in the blood of amyloid and phosphorylated tau proteins are associated with the corresponding CSF levels and amyloid-PET, tau-PET scans, and FDG PET [77,78,86,87]. Initially, most studies focused on the correlation between plasma Aβ and clinical or pathologic aspects of AD [78,86,87]. In addition, plasma oligomer Aβ (OAβ) negatively correlated with CSF Aβ42 and positively with CSF Tau but did not correlate with CSF p-tau levels. Combining plasma OAβ with APOEε4 and age has accurately identified brain amyloidosis, suggesting that using plasma OAβ as a pre-screen for amyloid PET would reduce the cost of PET scans [78]. To ensure the correlation between plasma Aβ levels and brain amyloidosis, high-precision assays to measure the plasma Aβ are necessary [77,88]. However, plasma or serum concentrations of Aβ42 and Aβ40 have not always been consistently associated with the clinical diagnosis of AD and Aβ PET deposition or tau PET (for a systematic review, see refs. [67,89]). Since tau is another valid candidate plasma biomarker for AD pathology, recent studies have shifted focus to plasma tau proteins [89]. Although blood measurement of p-tau is difficult due to its low concentration, substantial efforts show promise. Plasma p-tau181 could be a more specific marker, being exclusively of brain origin, than T-tau for the AD-specific pattern of amyloid PET and tau PET, since T-tau can have both cerebral and peripheral origins [89]. Plasma p-tau181 is currently considered an AD biomarker for execution in clinical practice [72]. Plasma p-tau181 significantly correlated with brain amyloid, tau PET [89], and FDG PET [72,90]. Moreover, a combination of plasma p-tau181 with amyloid was significantly correlated with tau deposition in the brain [72,91]. However, lack of correlation between blood-based and ATN biomarkers can be caused by several factors (e.g., variability between laboratories and batches of commercial assay, different stages of the disease, and the different reference standards to assess the performance of blood-based tests). In fact, further evidence of the association is needed to validate blood-based biomarkers for clinical practice. Additionally, well-correlated blood-based and core CSF biomarkers or brain imaging will robustly ensure the AD diagnosis and prognosis. This is an essential approach to developing the roadmap to the utility of blood-based biomarkers as an alternative to invasive and costly CSF biomarkers and the radioactive burden of brain imaging. Of note, the reference standard of core AD biomarkers for comparison of blood-based biomarkers are different for each study; therefore, the standard reference should be validated.

### 4.2. Other AD-Associated Protein Biomarkers (Non-Aβ and Tau)

Neurofilament light chain (NfL) has been classified as an alternative N in the ATN framework [93]. Recent developments have allowed for measuring NfL concentrations in blood samples. Elevated plasma NfL can anticipate brain neurodegeneration, which increases and correlates with future atrophy, hypometabolism, and cognitive decline. Thus, plasma NfL might be considered a blood-based biomarker for screening and tracking neurodegeneration in AD [94,95,96]. It should be noted that increased peripheral NfL in the blood and CSF are nonspecific to disease etiology, and studies of the precise mechanisms underlying their release and trafficking from CNS to peripheral blood are still lacking [97]. Glial fibrillary acidic protein (GFAP) is one of the AD biomarkers and plays a role in astrocytic activation and degeneration. Plasma or serum GFAP concentrations are elevated in individuals within the clinical AD continuum [98,99,100]. While GFAP levels in blood and CSF are associated and correlated to amyloid pathology [99] and related to clinical disease severity [100], GFAP might be a potential biomarker of other types of dementia because astrocytes are not the foremost cell type specific to AD pathophysiology. Astrocytic damage might begin during the pre-symptomatic stage of AD and is associated with amyloid load. Evidence from Pereira et al. confirmed that plasma GFAP levels were strongly associated with brain amyloid-β pathology but not tau aggregation [101]. It could be considered as a widely available screening tool to identify astrocytosis in early AD [101]. Therefore, GFAP and amyloid pathology levels may potentially differentiate A+ and A− in the ATN approach. In addition, Verberk et al. proposed using NfL or GFAP to measure “N” since they exemplify neurodegeneration [99]. Of note, both plasma NfL and GFAP are strongly associated with brain-specific products, suggesting that they may serve as good markers of CNS diseases. Neurogranin (NGRN) is a postsynaptic protein expressed primarily on the cerebral cortex, hippocampus, amygdala, and striatum. It has been recently proposed as a promising biomarker of synaptic dysfunction, especially in AD [102]. However, these non-amyloid biomarkers are not specific to AD, and therefore these biomarkers are likely adjunctive to core AD biomarkers for the determination of amyloid pathology or neurodegeneration.

### 4.3. Exosome

Neuronal-derived extracellular vesicles (e.g., exosomes) were isolated from the blood of AD patients and proposed as a blood AD biomarker [103,104,105]. Circulating exosomal contents (i.e., mRNA, protein, miRNA, and lipid) in blood as AD biomarkers gained attention and showed promising results initially. For instance, amyloid, tau proteins, and autolysosomal proteins that were extracted from neurally derived blood exosomes showed an ability to predict AD development in the preclinical stage [106,107]. Numerous studies of various miRNAs in extracellular vesicles (EVs) of plasma or serum showed the intriguing development of AD biomarkers [104,108]. The systematic review and meta-analysis showed that exosome-derived AD biomarkers showed potential as biomarkers for AD and MCI [109]. However, the communication protocols are difficult to follow, several hurdles need to be overcome, and the directions for the isolation of blood-derived exosomes for AD biomarkers are challenging. In fact, the lack of a standardized method for the collection and preparation of biological fluid, an optimized EV isolation method with high purity and yield from biologic fluids, and good analytical performance for the analysis of EV biomolecules (e.g., miRNAs and proteins) should be elucidated for the clinical application of EV biomarkers. For example, the exosome isolation procedure typically uses a commercially available kit based on precipitation using a polymer which may not be specific enough for exosome-specific biomarkers [104]. In addition, broad and novel contents in exosomes are found by methodological and analytical differences between studies. Thus, to become a potential biomarker for AD in clinical practice, further studies with standardized study designs for replication and validation require countless endeavors.

### 4.4. MicroRNA

MicroRNAs (miRNAs) have recently emerged as promising cost-effective and non-invasive biomarkers for AD, since they can be readily detected in different biofluids, particularly in the blood (as free form or within exosomes) [110]. Several investigations indicated potential miRNAs as biomarkers in blood (e.g., plasma and serum) for AD and MCI [111,112] and to discriminate between AD and non-AD patients with high sensitivity and specificity [113,114]. For example, miR-125b, miR-455-3p, and miR-501-3p have been suggested as having the potential to separate AD patients from healthy individuals with high sensitivity and specificity [113,115,116]. In addition, the combination of miRNA as a panel in blood has been introduced; for example, 12-miRNA showed 93.3% accuracy to separate AD patients from control [114]. A recent systematic review suggested that circulating miRNAs represented an excellent biomarker for the clinical diagnosis of AD [117,118]. Although miRNAs in the peripheral circulation closely reflect AD pathophysiology and serve as an ideal AD biomarker, they have recently been found to be associated with other disease pathologies (e.g., miR-125b and miR-455-3p involved in cancer [119,120]), and inconsistencies exist between different studies (e.g., miR-210-3p [118,121]). The limitations of miRNA discoveries using library construction followed by next-generation sequencing include the relatively low amount of total RNA from a blood sample, biased miRNA library construction, and the variability of sample processing, which might cause unreliable results [122,123]. Unlike core AD biomarkers, blood miRNA biomarkers also require a longitudinal study to identify common and reproducible characteristics before becoming a mature AD biomarker, which will provide clinical validity.

### 4.5. Lipids

Several lipid candidates have been introduced and showed a strong association with gold standard AD biomarkers (e.g., CSF p-tau/Aβ42 ratio). A systematic review shows that plasma cholesterol levels, triglycerides, sphingolipids, phospholipids, and various cholesterol derivatives are altered in AD [124]. Additionally, alteration of lipids in blood can be observed at the earliest AD phase that might provide insight into the biochemical processes engaged in AD pathogenesis. Studies of blood metabolism highlighted the role of lipid compounds, (e.g., phosphatidylcholines (PCs), phosphatidylinositols (PIs), and phosphatidylethanolamine (PE)) in AD [125,126,127]. It has been reported that the ratio of phosphatidylcholines to lysophosphatidylcholines in plasma can differentiate healthy controls from AD and MCI. The studies of plasma lipids for AD biomarkers are at an early stage and there are several controversies. For example, Mapstone et al. described a panel of 10 lipids from blood that can predict phenoconversion to MCI or AD within 2–3 years with 90% accuracy [125]. A year after that, Fiandaca reported another 10 plasma lipid panels that can predict phenoconversion with 90% sensitivity and 85% specificity [106]. However, the pathology behind those lipids (e.g., cholesterol, sphingolipids, phospholipids) and the origin of the brain and peripheral lipids needed further intensive studies. Combining several lipids as a panel may have advantages over a single lipid measurement.

### 4.6. Genetic Biomarkers

Among genetic variants associated with sporadic AD, APOE ε4 allele is the best-established genetic risk factor. In contrast, the presence of ε2 allele is associated with protective role of ApoE in AD, and ε3 allele is neutral. In addition, APOE ε4 allele plays a role as a modulator of the relationship between plasm Aβ and amyloid plaque in the brain. APOE ε4 negative subject but not ε4 positive subjects showed a positive correlation between plasma Aβ42 level and Aβ accumulation assessed by Aβ-PET [128]. The expression pattern of plasma proteins measured by a multiplex immunoassay was differentiated by the APOE genotype, which may indicate the importance of the APOE genotype in the analysis of blood biomarkers. Using next-generation sequencing, recent GWAS studies or meta-analysis suggested additional genetic variants associated with AD. For example, genetic variants of ABCA7, BIN1, CD33, CLU, CR1, EPHA1, PICALM and TREM2 genes were significantly associated with the progression of AD [129]. The roles of these genetic variants in AD pathogenesis and progression are largely unknown; however, the function of the genotypes in clinical classification using ATN framework may be significant. For example, APOE genotypes are associated with Aβ deposition (A) rather than tau pathology (T), while TREM2 genotypes are associated with tau pathology and stage dependence rather than Aβ deposition [130].

## 5. Perspective and Future Directions

During the past two decades, it has been found that the development the CSF AD biomarkers might be useful for early diagnosis, even in situations where there is a lack of disease-modifying therapies, providing an opportunity to start disease-specific care and the monitoring of the disease earlier. The use of valid CSF biomarkers for the early discrimination of patients in AD continuum from healthy controls or patients with non-Alzheimer-type dementia can be integrated into the design of therapeutic clinical trials, which will have significant value for increasing the efficiency or probability of success in the development of disease-modifying therapies. In addition, considering clinical practice, the use of biomarkers for ATN classification of patients may provide the expected prognosis to the clinician and care giver. Nevertheless, the addition of non-invasive and cheap blood analysis to the current valid AD biomarkers (i.e., CSF analysis or imaging biomarkers) may improve the diagnostic accuracy and ability to repeated monitoring the biologic changes. There has also been advancement in imaging technologies with several clinical implications and rapid improvement in the field of AD. The development of valid biochemical and imaging biomarkers has clear biological and clinical significance. First, CSF and imaging biomarkers allow us to define AD biology and show the temporal evolution of biomarkers over the disease’s entire progression in patients who develop AD. These advances make it possible for CSF AD biomarkers and PET imaging to define the AD continuum’s antemortem characteristics more accurately compared with clinical diagnosis. Second, considering the barriers to applying universal cut-off values of CSF AD biomarkers for early AD diagnosis, the current technical advances in fully automated analytical platforms that significantly reduce the height of obstacles are expected to make CSF biomarkers usable in clinical practice. In addition, the evolution of AD biomarkers has led to the integration of biomarkers as ATN schemes that are beneficial for clinical practice and research for defining pathologic status over the AD continuum and discriminating from non-AD dementia. Third, novel non-amyloid CSF or blood AD biomarkers are considered next-generation biomarkers for AD, although further validation and replication are both necessary. Notably, the blood-based biomarkers approach presents several advantages that may map the road to clinical application. Additionally, amyloid or tau-based approaches may lead to the discovery of novel therapeutics. These novel biomarkers will provide insights to develop targets for AD therapeutics and facilitate the development of disease-modifying therapy and preventive medicine. Lastly, biomarker discovery and development with continuous progress in analytical performance could facilitate clinical trials and optimize the biomarker-guided strategies by integrating biomarkers into clinical trial design and use for AD diagnosis in clinical practice.

## Figures and Tables

**Figure 1 biomedicines-10-00850-f001:**
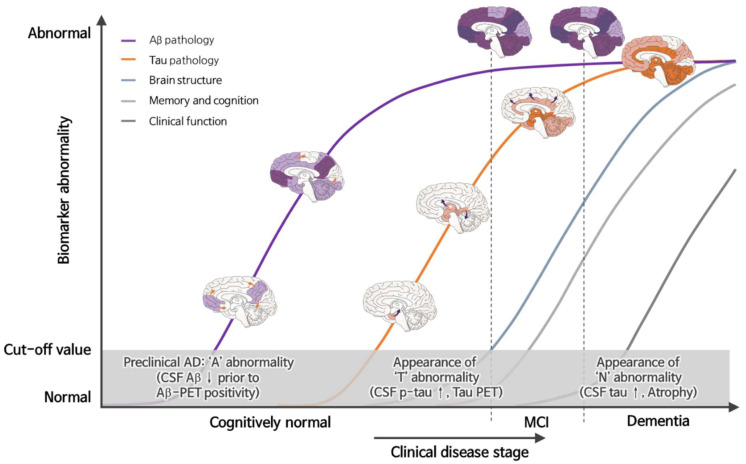
Biomarkers and changes in spatiotemporal pattern of Aβ and NFT deposit during disease cascade. Abnormality in CSF Aβ42 and Aβ-PET (‘A’ abnormality) are biomarkers in pre-clinical stage, followed by abnormality in CSF p-tau or Tau-PET and ‘N’ abnormality (increased CSF T-tau and hippocampal atrophy).

**Figure 2 biomedicines-10-00850-f002:**
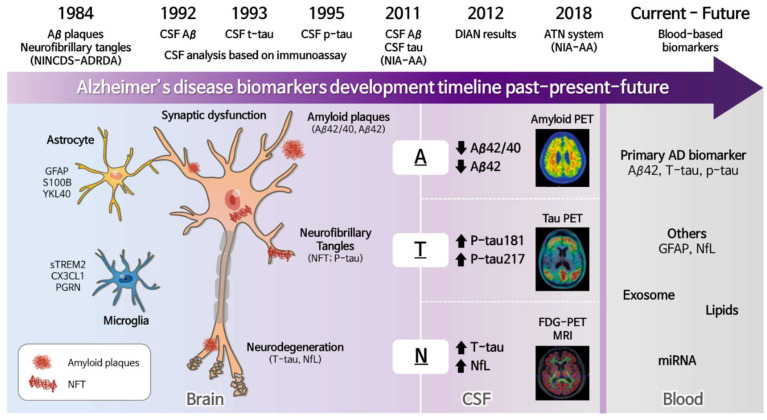
Schematic summarizes the milestones of Alzheimer’s disease biomarkers evolute during the past century. In 1984, the first release of criteria for the clinical diagnosis of AD (NINCDS-ADRDA criteria) was published, and a definitive diagnosis of AD can be achieved only by the detection of Aβ plaques and NFT by autopsy. Subsequently, brain imaging technologies to detect amyloid accumulation in the brain using various PET ligands bound to amyloid plaque have emerged, and more recently tau PET has been introduced. Later, in 2011, the NIA-AA recommended diagnostic guidelines for non-symptomatic (i.e., preclinical) and symptomatic (i.e., MCI and dementia) stages. The recent new NIA-AA, updated in 2018, revised the diagnostic criteria for research and provided the new diagnostic approach based on the ATN classification system for the research framework. There are several molecular changes from neuron and glial cells (e.g., astrocyte and microglia) associated with CSF and/or blood biomarkers. If clinical evidence of the glial biomarkers for AD diagnosis becomes valid, the glial cell-derived biomarkers can contribute to clarifying the ATN classification in the future. Up-to-date blood-based biomarkers have been focused on and promised as new AD biomarkers. The PET images from [36,37] are used with permission from the publishers. NFT; neurofibrillary tangles, GFAP; Glial fibrillary acidic protein, CSF; Cerebrospinal fluid, PET; positron emission tomography; FDG-PET; fluorodeoxyglucose positron emission tomography, MCI; mild cognitive impairment, NIA-AA; National Institute on Aging and the Alzheimer’s Association.

**Table 1 biomedicines-10-00850-t001:** The correlation between plasma blood biomarkers and core CSF biomarkers or brain imaging.

Aβ Biomarkers (A)	Analysis Method	Sample Size	Correlation ith Reference Standard	References
Plasma Aβ42/Aβ40	SIMOA	*n* = 719	Negative correlation Aβ PET SUVR Positive correlation CSF Aβ42	[87]
Plasma Aβ42/40	High-precision immunoprecipitation mass spectrometry (IPMS)	*n* = 158 (Cognitively normal)	Negative correlation Aβ PET positivity (AUC 0.88, 95% CI 0.82–0.93) CSF p-tau181/Aβ42 (AUC 0.85, 95% CI 0.79–0.92)	[77]
Plasma Aβ42/Aβ40	ELISA (Aβtest, TP42/40)	*n* = 135 (18-month) *n* = 169 (36-month) *n* = 135 (54-month)	Negative correlation Aβ PET SUVR (r = −0.63, *p* < 0.0001)	[86]
Plasma Aβ42/Aβ40	Liquid chromatography-tandem mass spectrometry (LC-MS/MS)	*n* = 414	Negative correlation Aβ PET positivity (AUC 0.81, 95% CI 0.77–0.85)	[75]
Plasma Aβ Oligomer	Multimer detection system (MDS)	*n* = 399 (MCI *n* = 42) (AD dementia *n* = 164) (non-AD dementia *n* = 58) (Other disease *n* = 61) (Normal control *n* = 60) (Subjective cognitive decline *n* = 14)	Positive correlation CSF T-tau (r = 0.20, *p* = 0.01) Negative correlation CSF Aβ (r = −0.20, *p* = 0.035) No correlation p-tau (r = 0.12, *p* > 0.05)	[78]
Plasma Aβ42/Aβ40	High-precision immunoprecipitation mass spectrometry (IPMS)	*n* = 465	Negative correlation Aβ PET positivity (AUC 0.84, 95% CI 0.80–0.87) Positive correlation CSF Aβ42/Aβ40 (AUC 0.85, 95% CI 0.78–0.91)	[88]
**p-tau Biomarkers (T)**	**Analysis Method**	**Sample Size**	**Correlation**	**References**
Plasma p-tau181	SIMOA (* substituting the detection antibody for a p-tau 181-specific monoclonal antibody)	Cognitively unimpaired; *n* = 172, MCI; *n* = 57, AD dementia; *n* = 40	Positive correlation Tau PET SUVR (p-tau181; r = 0.580, *p* < 0.001)	[89]
Plasma p-tau217 Plasma p-tau181	Electrochemiluminescence-based assays (different in the biotinylated antibody epitope)	*n* = 593	Positive correlation Aβ PET positivity (p-tau217: AUC = 0.91, 95% CI = 0.88–0.94, p-tau181 AUC = 0.89, 95% CI = 0.86–0.93)	[91]
Plasma p-tau217 Plasma p-tau181	SIMOA	Subgroup of 40 subjects with Aβ PET (*n* = 300) Autopsied sample *n* = 113	Positive correlation Aβ PET positivity (p-tau217: AUC = 0.84, 95% CI = 0.68–0.99, p-tau181: AUC = 0.82, 95% CI = 0.65–0.99) Presence of AD pathology (*p* < 0.001)	[84]
Plasma p-tau181	SIMOA	*n* = 1189	Positive correlation Aβ PET SUVR (r = 0.45, *p* < 0.0001) Tau PET SUVR (r = 0.25 *p* = 0.0003) Negative correlation FGD PET uptake (r = −0.37, *p* < 0.0001)	[72]
Plasma p-tau217	Meso Scale Discovery-based immunoassays	*n* = 490 Cognitively health control *n* = 225; Subjective cognitive decline *n* = 89 MCI *n* = 176	Positive correlation CSF p-tau217 (r = 0.709 in Aβ-PET positive Control; r = 0.543 in Aβ-PET positive MCI) Entorhinal Tau PET (87% agreement)	[92]
**Neuronal Injury Biomarkers (N)**	**Analysis Method**	**Sample Size**	**Correlation**	**References**
Plasma T-tau	SIMOA	*n* = 97 (Normal control *n* = 68) (AD *n* = 29)	Poor correlation CSF T-tau (r = 0.26, *p* = 0.09) Positive correlation CSF p-tau 181 (r = 0.29, *p* = 0.003)	[79]
Plasma T-tau	SIMOA	Cognitively unimpaired; *n* = 172, MCI; *n* = 57, AD dementia; *n* = 40	Positive correlation Tau PET SUVR (T-tau; r = 0.194, *p* = 0.022)	[89]
NfL *	SIMOA	Autopsied samples (*n* = 113)	Positive correlation Presence of AD pathology (*p* = 0.07)	[84]

* NfL: Neurofilament light.

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
