# Peer review of "Development of Alzheimer’s Disease Biomarkers: From CSF- to Blood-Based Biomarkers"

_biomedicines, 2022, doi:10.3390/biomedicines10040850_

Round 1

Reviewer 1 Report

Mankhong et al. reviewed and discussed different groups of blood-based biomarker candidates of AD therapeutics and diagnostics. Overall, this review article is very well-written and easy to follow. However, there are a few suggestions for authors to supplement and complete the story.

  1. Before “2. Discovery and development of AD biomarkers”, authors should add (i) a graph showing clinical disease stage of AD, including pre-symptomatic, mild-cognitive impairment, late-cognitive impairment, and dementia, versus disease hallmarks at different stages. Also, (ii) a spatiotemporal pattern of Ab and neurofibrillary tangle deposit during the disease cascade in human AD brain should provide. These two additional illustrations will point out essential roles of early biomarkers in AD.

  1. Authors should provide details about genetic risk factors of AD, such as APOE4, TREM2, that use with ATN biomarker profiles and clinical diagnosis.

  1. Authors put astrocyte and microglia in Figure 1. Should they provide what biomarkers on these cell types in this figure and figure legend? In addition, they should discuss the details in the text regarding glial cell biomarkers for AD.

  1. Several lines of evidence suggest that soluble tau oligomers are the most toxic form and detect in CSF samples in AD patients. Additionally, specific post-translational modifications of CSF-Tau have been reported. Can authors discuss about these points?

  1. Recent studies have shown that other proteinopathies, such as a-synuclein and TDP-43, were detected in AD brains at late stages. Please further discuss on this point linked to the aspect of AD biomarkers.

Author Response

We appreciate the helpful reviewer's critical comments to improve our manuscript. In revised version, we remarked the edited sentences as red-colored text. In the file of response, we responded to the comments in the point-by-point manner.

Reviewer 2 Report

In the last few years the role of biomarkers for the diagnosis of Alzheimer disease (AD) became fundamental thanks to the he development of optimal immunoassays for measuring AD biomarkers in humans particularly for measurement of CSF A1-42, 416 p-tau, and t-tau. In the present review the authors explore the past, the present and the future of AD biomarkers in depth. 

According to the authors CSF and imaging biomarkers allow us to define AD biology and show the temporal evolution of biomarkers over the disease’s entire progression in patients who develop AD and to define the AD continuum’s antemortem characteristics more accurately compared with clinical diagnosis.

The review is well written but some informations are still missing, and I ask the authors to address  better the following issues:

  • The interpretation of a single biomarker positivity (eg. isolated amyloid reduction or isolated tau increment).
  • The role of biomarkers in other diseases such as Lewy body disease or fronto temporal dementia
  • The role of Aβ40 and Aβ42 in AD pathophysiology and their ratio as possible diagnostic tool in the cerebrospinal fluid

Author Response

We appreciate the reviewer's helpful comments to improve the quality of our manuscript. In the revised version of manuscript, we remarked the edited sentences by red-colored text. In the attached file of responses, we responded to the comments in a point-by-point manner.

Reviewer 3 Report

This review by Mankhong et al. is at first sight an in-depth review of blood and CSF biomarkers of Alzheimer´s disease. The authors know their subject and present an up-to-date reference list. The paper is not very novel, however; several of the cited reviews present similar material (cf. Refs. 7, 32, 36-39, 60).

On second thought, however, several weaknesses come to mind that require the attention of the authors:

  1. The paper is wordy. The first 4 pages are too long with only a few meaningful points. The paper has many repetitions of ideas in other paragraphs, eg. the advantages of automated immunoassay (line 141, 172) or the correlation of biomarkers.
  2. Point 4.1 and Point 5 can probably be combined, they cover similar ground.
  3. A table about the differences/similarities of the analytical methods (eg. LOD, LOQ, analyte, price/sample, limitation, ability to differentiate the central produced vs peripheral produced biomarkers) may be more interesting than a wordy paragraph.
  4. The review is often quite descriptive. Throughout, the authors give little information about the precision and specificity of the procedures. Using the present method, how many patients would be missed (1st order error) and how many would be falsely diagnosed (2nd order error)? This information is crucial because large-scale use of the suggested parameters would identify more false positives than patients with AD and would do more harm than good.
  5. Along the same lines, the authors should comment on the usefulness of a diagnostic procedure if there is no therapeutic value. In other words, can you justify the withdrawal of CSF (a potentially painful and dangerous procedure) in a possible AD patient if there is no prescription drug that stops the disease? What is the value of such procedures?
  6. On p. 4 ff., the authors should explain in more detail what the different procedures do. As it stands, the review is fine for a reader who is already well informed but rather useless for a non-specialist because the value of FDG-PET or MRI is poorly explained. It may be argued that neuronal cell death is a more important parameter than amyloid plaques but do the PET and MRI techniques have predictive value?
  7. On p. 7, GFAP is likely a marker of astrogliosis (neuroinflammation) which is not specific to AD and neurogranin is probably increased after stroke. Why neurogranin should be an indicator of AD escapes the reader.
  8. The review does not discuss familial and spontaneous AD separately. It would be expected that familial AD cases had high Aß whereas the assays may be less specific in spontaneous AD cases. What about APOE4 in this respect?
  9. The review also suffers from inconsistent argumentation, especially in the later parts 4.3, 4.4. and 4.5. The authors do not seem to like these procedures, but they start the paragraphs by citing supporting evidence, then they list weaknesses (which is good) – but the weaknesses are poorly elaborated. In exosome research, “several hurdles need to be overcome” – which hurdles? The sentence in lines 333-334 is not understandable. miRNA “reflect AD pathophysiology … in various aspects”. Which aspects? While I share much of the skepticisms of the authors towards these procedures, a critical review should clearly point out the (present) weaknesses of these procedures and why they are (at present) not useful.
  10. Table 1 is useful but some information is missing in the columns (samples sizes, correlations etc.; please define NFL in the legend).

Author Response

We appreciate the reviewer's extensive and critical comments that are helpful to improve the quality of our manuscript. In the revised version of manuscript, we remarked the edited sentences by red-colored text and updated references as appropriate. In the attached file of responses, we responded to the comments in a point-by-point manner. 
